# Getting Rid of the Usability/Security Trade-Off: A Behavioral Approach

**Francesco Di Nocera *** and **Giorgia Tempestini**

Department of Planning, Design and Technology of Architecture—Sapienza University of Rome, 00196 Rome, Italy; giorgia.tempestini@uniroma1.it
* Correspondence: francesco.dinocera@uniroma1.it

**Abstract:** The usability/security trade-off indicates the inversely proportional relationship that seems to exist between usability and security. The more secure the systems, the less usable they will be. On the contrary, more usable systems will be less secure. So far, attempts to reduce the gap between usability and security have been unsuccessful. In this paper, we offer a theoretical perspective to exploit this tradeoff rather than fight it, as well as a practical approach to the use of contextual improvements in system usability to reward secure behavior. The theoretical perspective, based on the concept of reinforcement, has been successfully applied to several domains, and there is no reason to believe that the cybersecurity domain will represent an exception. Although the purpose of this article is to devise a research agenda, we also provide an example based on a single-case study where we apply the rationale underlying our proposal in a laboratory experiment.

**Keywords:** usability; cybersecurity; behavior analysis; token economy; gamification

## 1. Introduction

A leitmotif of the cybersecurity literature is the inversely proportional relationship between usability and security: more secure systems will necessarily be less usable (or, simply, less easy to use), and usable systems will be more vulnerable to threats [1,2]. While organizations lean toward more secure systems and practices, users prefer systems and practices that provide more usability, giving up on adopting a secure approach and exposing themselves to more risk, even when they are aware of existing threats [3].

The first concept to clarify is what is meant by usability. An official definition of usability is provided by ISO 9241:2018, which defines it as "the extent to which a product can be used by specific users to achieve specific goals with effectiveness, efficiency, and satisfaction in a given context of use" [4]. Many efforts have been made in the last half-century to improve the relationship between user and technology, to make the use of a device effortless and reduce the learning curve. With the spread of procedures that can be performed online, the need to develop user-centered technologies has increased exponentially. Today, it is common to use the Internet to perform tasks that were previously done in specific contexts, mainly outside the cyberspace. People are now purchasing services and products, obtaining information, and communicating through applications on personal devices. More than half of the world's population now uses the internet daily for numerous business and leisure activities [5], and the opportunities for cybercrime are naturally proportional to the prevalence of online activities. Cybercrime is steadily increasing, with a global annual cost of $600 billion [6]. An immediate example of an intervention to improve individual–technology interaction involving cybersecurity is saving login credentials with the aim to not have to remember them and enter the system without typing them. In the face of increased usability, this practice exposes to the likelihood that a malicious user gains access to the system and the library of saved passwords. Conversely, procedures such as creating complicated passwords or changing them constantly increase security

but make the human-system interaction more demanding. In addition, it leads the user to adopt security-violating practices, such as writing the password on a post-it note pasted on the computer.

The problem, then, is how to combine security with procedures and interfaces that can ensure a better user experience. Before addressing some specific hypotheses on how this can be achieved, the following sections will briefly discuss (1) the topic of cybersecurity, (2) some of the most important threats, and (3) the topic of "usable security".

## 2. Cybersecurity

Cybersecurity is the "protection of confidentiality, integrity, and availability of information in cyberspace" [7]. Specifically, it represents that part of information security that focuses mainly on protecting digital information against any threats that may arise from using the internet [8]. To understand more clearly what is meant by cybersecurity, it is appropriate to analyze the two terms "cyber" and "security". On one hand, "cyber" is a prefix that connotes cyberspace and refers to "a complex environment resulting from the interaction of people, software, internet, supported by physical information and communication technologies" [7]. On the other hand, concerning the term "security", a specific one has not been suggested in the literature [9], but it generally refers to something free of dangers or threats [10].

The attention given to cybersecurity is continuously expanding [11], especially due to the spread of the internet [8]. The attention paid to cybersecurity has also increased due to the global COVID-19 pandemic [12]. Organizations and companies have had to implement work from home (WFH) practices, leading workers to use devices (sometimes even personal devices) and connections lacking the standard security measures provided by companies, which has led to an increase in cyber-attacks and risks to corporate data [13].

A cyber-attack consists of any action taken to undermine the operation of a computer network to compromise political and national security [14]. It is, therefore, a disruption of normal computer operation and loss of information through a series of malicious events [15]. Cyber-attacks are implemented to spread misinformation, block certain services, access sensitive information, conduct espionage, steal data, and cause financial losses. As the last decade has seen an evolution of social media, this has led to the creation of the so-called "social cyber-attack", which can be of two different types: "(1) pre-meditated, which are designed to create an excited signal in a social network, often under false pretenses, so as to benefit from the chaos and upheaval; and (2) opportunistic, which take advantage of an existing excited social network signal and, by manipulating it through various means, derive benefit" [16] (p. 5).

The complexity and severity of cyber-attacks has increased over the years. There are still some gaps concerning the different types of attacks, which make some countries or some organizations particularly vulnerable [17].

Since the beginning of the global COVID-19 pandemic, in addition to the primary threat to the health of individuals around the world, there has been a dramatic increase in cyber-attacks as e-commerce increases [18]. Such attacks have generated harmful consequences for society. Data show that as in March 2020, there was a 600% increase in phishing emails (fraudulent messages created to appear authentic, requiring you to provide sensitive personal information) [19] and that, in April 2020, nearly 18 million malware and phishing emails were blocked every day [20].

The 2021 annual report on cyber threats published by the European Union Agency for Cybersecurity [21] identified the following as prime threats:

- Malware: more commonly known as "computer virus" malware (short for malicious software), means any computer program used to disrupt the operations performed by the user of a computer.
- Ransomware: a type of malware that restricts access to the device it infects, demanding a ransom to remove the restriction. For example, some forms of ransomware lock the

system and require the user to pay to unlock the system, while others encrypt the user's files and require payment to return the encrypted files to plaintext.

- Crypto-jacking: a computer crime that involves the unauthorized use of users' devices (computers, tablets, servers, smartphones) to produce cryptocurrency. Like many forms of cybercrime, the main reason behind them is profit. Unlike other threats, it is designed to remain completely hidden from the victim.
- Email-related threats: in this category of attacks, we find spoofing, spam, spear phishing, Business Email Compromise (BEC), whaling, smishing, and vishing. All these attacks the same characteristics concerning the exploitation of the weaknesses of human behavior, human habits, and the vulnerability of computer systems to push individuals to become victims of an attack.
- Threats against data: this category includes attacks where a data breach or loss occurs, and sensitive and confidential data ends up in an unprotected/secure environment. Taking over other people's data is certainly one of the main goals of hackers for many reasons, such as ransomware, defamation, extortion, etc. This type of breach can present in several ways: they can occur due to a deliberate cyber-attack or can involve personal and sensitive data being spread incidentally.
- Threats against availability and integrity: these attacks aim to make information, services, or other relevant resources inaccessible by interrupting the service or overloading the network infrastructure.
- Disinformation and misinformation campaigns: the main difference between these two types is that, the first case refers to the diffusion of false information to intentionally deceive people while; the second case concerns with the dissemination of of misinformation, misleading, inaccurate, or false information is provided without the explicit intention to deceive the reader. These campaigns reduce the general perception of trust and lead people to doubt the veracity of information.
- Non-malicious threats: a malicious user uses authorized software, applications, and protocols to perform malicious activities. This refers to the kind of threat in which the malicious intent is not evident, and the control of the infected device takes place without the need to download malicious files.
- Supply-chain attacks: this involves damaging the weakest elements of the supply chain. The goal is to access source code to create or update mechanisms, infecting apps to spread malware.

## 3. Usable Security

When employing the term "usability", it is difficult to avoid falling back on expressions such as "ease of use", "simplicity", or "intuitiveness". However, the concept fails to be captured exclusively by these terms. Although we are often content to define usability as ease of learning and using an artifact, there are many ways in which the quality of user–technology interaction can be described and measured.

The ISO 9241 standard, introduced by the International Organization for Standardization in 1998 (and revised in 2018), describes "usability" as the degree to which specific users can use an artifact to achieve certain goals with effectiveness, efficiency, and satisfaction in a specific context of use [4].

Typically, effectiveness coincides with the achievement of goals, efficiency with the time it takes to perform a task, and satisfaction with users' subjective ratings. These are, of course, simplifications of a more complex concept. However, although this "standard" definition represents a compromise between different theoretical instances, it does not provide clear operational indications. The standard highlights an important aspect that is often overlooked by different theoretical and procedural approaches: usability is not a characteristic of the product itself but depends on the characteristics of the users of the system, on the goal that they intend to achieve, and on the context in which the product is used. Implicitly, the standard underlines how usability cannot be traced back to the presence/absence of characteristics but must be assessed considering the individual's

subjective experience. With their abilities and limitations, users intend to achieve an objective using technology and wish to do so using the least number of resources possible and not having to invest more, paradoxically, because of the technology they use. Design, therefore, cannot disregard the knowledge of the users' needs, limits, and potentialities and the careful task analysis that will have to be carried out while using a device in each context.

Information technology security also needs to be usable; the expression "usable security" indicates managing security information in the user interface design. In this context, usability is fundamental at different levels: (1) from the user's point of view, because it allows completing a task effectively, efficiently, and satisfactorily, avoiding errors that may cause security problems; (2) from the developer's point of view, as it is crucial for the success of a system; (3) from the management's point of view, considering that weak security mechanisms could represent a limitation to the usability of the system [22].

Security, then, is not a functionality divorced from usability but is related to it, and the designer's goal must be to ensure both security and usability while preventing one from compromising the other [22]. Furnell [23] (p.278) pointed out that "the presentation and usability of security features, in some cases, are less than optimal", requiring effort from the user to use them correctly. This effort is necessary but, in some cases, reduces the usability of a procedure and, therefore, discourages its use. Gunson and colleagues [24] reached the same conclusion when comparing the usability of two authentication methods in automated telephone banking. They conducted empirical research involving customers of a bank. The objective of the study was to compare the level of usability perceived by users using one-factor and two-factor authentication methods. Two-factor authentication is a security procedure in which users provide two different authentication factors to verify their identity.

The results confirmed how participants considered the two-factor authentication less easy to use but, at the same time, they were aware that it was the most secure method [24]. This example precisely describes the problem of managing usability and security, with the two components often being at odds. The consequences of mismanagement of this trade-off could result in procedures that are either at high risk of violation or, conversely, too complex. In the latter case, the excessive effort to engage in secure behaviors could lead to the emergence of less secure alternative behaviors [25].

Within companies and organizations, all these processes, which ensure information security, are designed according to a top-down perspective, generating security policies that are not centered on the end-users. Therefore, they obtain the paradoxical effect of being ignored by users who, while adhering to the policies, would perform their tasks with difficulty. In contrast, a desirable approach to ensuring usable security should be to understand the user's behaviors to generate viable security practices [26,27]. For example, Bravo-Lillo and colleagues studied the effect of habituation on security dialogue boxes, which leads users to ignore important messages due to repeated exposure to these dialogues.

The user reports reading them (by automatically pressing a button) but without paying attention to the content [28,29]. It seems clear that a "fulfillment" mode to the design of security features (i.e., security alerts must be presented, and the user must report having read them) does not work because it does not consider processes that involve the interaction between the system and the individual with their human characteristics, such as, in this case, the process of habituation to repeated stimulation.

One of the most studied topics in usable security is the problem of passwords, which are constantly between usability and security. Interfaces for password creation and entry are the most implemented authentication methods in modern systems. Hundreds of thousands of websites and applications require users to enter a password to access a system [30]. The goal of passwords is twofold: the first is the security goal, which requires passwords to be complex, unique, and difficult enough to not allow hackers to identify them; the second is the usability goal, which requires passwords to be easy enough for the user to remember them [31,32]. Nowadays, many online services request the adoption of passwords that have a certain degree of complexity (e.g., use at least one capital letter, a number, and a special

character and are at least eight characters long), and security policies in organizations often demand password replacement after a certain time interval.

Moreover, policies often require the user to use different passwords for different systems. They suggest not using the same password to access multiple services and storing the login credentials in a place inaccessible to others. Once again, this is an example of a top-down policy that is not based on a user-centered design rationale.

When users feel overwhelmed by the system demands, they could dismiss the system itself [25]. People seem to carry out a cost-benefit analysis associated with safe behaviors [33]. Users will avoid spending too much cognitive and temporal resources on security if the perceived benefits are too low [34]. Consequently, system usability becomes a critical factor when analyzing why people behave unsafely in cyberspace [35]. Indeed, while ease of use seems to be correlated with security weakness, more secure systems are difficult to use. The challenge is to bring together security and usability, which are usually perceived as mutually exclusive [22].

In contrast to other contexts where usability can be addressed as an independent goal when security solutions are developed, it is paramount that usability is evaluated in relation to it [36]. Mechanisms designed to ensure security should never restrict the user from performing the main task but should be designed to recognize human limitations and prevent users from dealing with unusable systems [37]. However, attempts to combine usability and security are often limited to improving the transparency of security processes; they do not make the system usable, but make the communication of information and procedures usable [38]. While this is an entirely reasonable and worthwhile goal, there remains the insurmountable problem of users' tendency to override security practices that are perceived as obstacles to the effectiveness, efficiency, and satisfaction of their interaction with the system. In this paper, we intend to propose an alternative approach to that adopted thus far in the literature: rather than trying to reduce the gap between usability and security, we suggest accepting the existence of the usability/security trade-off to ensure adherence to security procedures by compensating with increased usability. To this end, it is appropriate to briefly describe the theoretical assumptions of behavior analysis that will clarify the proposed intervention model.

## 4. Behavior Analysis: A Primer

Among the different perspectives adopted in psychology to explain individuals' behavior and make predictions, Behavior Analysis has a role of primary importance. The behavioral view (see [39] for an extended discussion) is based on the idea that consequences are crucial in producing repeated behaviors. A person who plays slot machines will play again if he or she has won on other occasions. So, when behavior is selected by its reinforcing consequences, its frequency of emission increases. Conversely, behavior that is not followed by a reinforcing consequence decreases in frequency, up to extinction. This process is called "operant conditioning", and it is the primary modality in which organisms modify their behavior throughout their life experience. The term "operant" indicates that a behavior operates on the environment to produce effects. We dial a phone number to communicate with the person we want to contact. Communicating with the person, in this case, is a reinforcer, and the consequence of the behavior is the fundamental element in defining the operant. For a behavior to be defined as an operant, it is not necessary to dial the phone number by typing the keys on the keyboard, asking Siri, or even asking another person to do it for me, if all these actions allow me to obtain the same effect. All these actions pertain to the same class of operants, even if they vary in their topography (i.e., in the form that the responses take). For this reason, we speak of response classes, which are united by the function they fulfill.

Taken as a whole, the events that signal the opportunity to enact a behavior (alternatively called discriminative stimuli), the class of operants, and the consequences that follow the operant behavior all constitute what is called the "reinforcement contingency" or "three-term contingency".

Discriminative stimuli have a paramount role in regulating operant responses. Still, they are not the cause of behavior; they are events that regulate behavior, because, in their presence, the latter has been previously reinforced. If the phone rings and we answer, our behavior does not depend on the fact that the phone has rung; rather, it happens because, on other occasions when the phone has rung, answering has put us in communication with the speaker. In this case, the consequence increases the likelihood of the behavior in the presence of the discriminative stimulus; therefore, we call this consequence a reinforcer.

Moreover, how reinforcement is delivered determines typical response patterns that are independent of the organism, the type of behavior, and the type of reinforcement used. Reinforcement schedules can be fixed-ratio (reinforcement is delivered after the production of a specific number of responses), variable-ratio (reinforcement is delivered after the production of a variable number of responses), fixed-interval (reinforcement is delivered after the production of a response at the end of a specific time interval), or variable-interval (reinforcement is delivered after the production of a response at the end of a variable time interval).

The simplest example of a reinforcement schedule is the Fixed Ratio 1 (or continuous reinforcement) schedule: each response is followed by a reinforcer. When we flip the switch to turn on the light, the light bulb turns on. Every time. In nature, reinforcements are often not continuous. Organisms engage behaviors that are occasionally reinforced, as reinforcement is intermittent; however, this very intermittence makes the production of the behavior particularly robust. The criterion that organisms adopt is "sooner or later it will work, as it has worked in the past".

How can behavior analysis be usefully employed to help individuals cope with cyber threats? First, we need to reframe the role of usability in determining the use of security tools and procedures.

We know that safety tools are necessary but not sufficient for creating a safe system. Indeed, as reported by Furnell, Bryant, and Phippen [40], even if a tool is available, it is not always implemented. For example, low levels of software updating (weekly from 37% to 63%) are observed, despite the high rate of software installation (93% with anti-virus software, 87% with firewalls, 77% with anti-spyware, and 60% with anti-spam), suggesting that human behavior is a crucial factor in prevention [41]. In this regard, in 2007, Global Security Survey reported "human error" as the most reported cause of failures of information systems [42]. The human component is also considered a relevant factor in more recent years. The IBM Security Service [43], for example, observed that 95% of security breaches were due to human error. El-Bably [44] investigated the behavior of employees of companies in the UK and the US, reporting that the percentage of employees who had committed errors concerning security processes was slightly lower than 50%.

Recently, a study by Kannison and Chan-Tin [45] found a relationship between psychological features and safety behaviors, therefore confirming the relevance of the human component in the implementation of safety behaviors.

Furnell [23] suggested that the correct use of a tool relies on the awareness of its usefulness: if users do not understand or are not aware of the security risks, they are more vulnerable to incorrect behaviors. Moreover, users may be aware of the risk but may not know the correct behavior. Indeed, the more complex the tool, based on concepts such as cryptography, access keys, and digital signature, the more of an obstacle it becomes so that people try to get around it. Password management is a clear example of this problem: a strong authentication method is highly demanding and increases user workload [46]; therefore, incorrect low-demanding behaviors are engaged instead. When the users feel overwhelmed by the system demands, they can dismiss the system itself [25]. As reported above, people seem to carry out a cost-benefit analysis associated with safe behaviors [33]. Users will avoid spending too much cognitive and temporal resources on security if the perceived benefits are too low [34]. Consequently, system usability becomes a critical factor to explain why people behave unsafely [46].

As we reported at the beginning of this paper, the security/usability trade-off cannot be avoided, but we suggest that it can be exploited to devise design strategies for the improvement of security. The behavioral perspective becomes crucial in this attempt. Recently, we have witnessed a renewed interest in the behaviorist model concerning understanding certain phenomena related to the use of digital technologies and the design of interfaces and procedures able to encourage the acquisition of new habits [47]. However, the marketing perspective with which these issues are approached has distanced itself from the conceptual and methodological rigor of behavior analysis, often jeopardizing fundamental concepts and creating dangerous makeup operations by using different terms to indicate things with a specific meaning (e.g., the use of the term reward to indicate reinforcement). This is not the place to address these issues. Still, it is important to emphasize that the complexity of the model underlying behavior analysis (or functional analysis) requires conceptual, terminological, and methodological rigor.

For example, there is much literature on the topic of "gamification" [48]: the use of game-like modalities to achieve learning objectives. The principle behind gamification is to use the dynamics and mechanics of gaming, such as accumulating points, achieving levels, obtaining rewards, and exhibiting badges. However, long before the term gamification made its way into the literature, this dynamic was well known (and still it is) under the name "token economy". The token economy is a rather complex reinforcement system based on the accumulation of objects, namely tokens that can be eventually exchanged for goods, services, or privileges (the "back-up" reinforcements). The operating principle of the token economy is similar to the monetary system; it is a set of rules that determines the value of an object without an intrinsic value (just like a coin or a banknote). As tokens become exchangeable, the value of terminal reinforcements and the schedules employed to obtain both tokens and back-up reinforcers constitute the variable the experimenter can manipulate.

Ayllon and Azrin [49] were the first to implement the token economy as a behavior modification strategy. The first study was conducted on psychiatric patients, and the token economy was used as a reinforcement system to modify a range of behaviors. Subsequent studies [50] have supported and accumulated data on the efficacy of the token economy by applying it to settings other than mental health [51,52]. Today, the token economy is the strategy chosen for the treatment of autism spectrum disorders [53].

The token economy consists of six primary elements [54,55]:

- Target behavior: the behavior required for receiving tokens. This behavior must be objective and measurable.
- Token conditioning: the procedure through which the token is conditioned as reinforcement.
- Back-up reinforcement selection: the method by which the activities that can be acquired through the token exchange are identified.
- Token production schedule: schedule of reinforcement through which tokens are released.
- Exchange production schedule: a schedule that defines when tokens can be exchanged for back-up reinforcement.
- Token exchange schedule: schedules that determine the cost of back-up reinforcement in terms of tokens.

Moreover, interventions based on the token economy allow assessing the effectiveness of the treatment at the level of the single individual. The effects that are reported in the psychological literature are almost always based on studies that examine average scores from measurements obtained on a high number of subjects and analyzed using techniques based on inferential statistics. Of course, there are different circumstances in which this is desirable, but single-subject designs [56] allow the examination of the time course of a phenomenon. Effects can be studied in more detail by comparing the baseline (i.e., how the subject behaved before treatment) with what happens after the introduction of the treatment and, subsequently, its removal. We believe that this research strategy may be more useful to better understand the dynamics of the interaction with security systems.

As an example, consider the following plot (Figure 1) showing the performance of a 23-year-old female subject in discriminating between suspicious and non-suspicious emails in an experimental study designed to be realistic but not real. The d-prime metric was used as the dependent variable. In Signal Detection Theory, the discriminability index (d-prime) is the separation between the means of the signal and the noise distributions. The task was to read 90 emails (30% of them were suspicious) and correctly categorize them. The "system" required the user to enter a password for each categorization. This was necessary to create discomfort for the subject. The subject performed the task for three days (Baseline), then entered the Treatment condition (three days), in which she received a bonus point for each correct identification (token). Whenever she correctly categorized 10 emails, she received access to the benefit (backup reinforcement) of not entering the password for the following 10 emails (and so on, until the end of the task). After three days of Treatment, the subject entered the Baseline condition again (no tokens collected, no back-up reinforcement for three more days). The plot shows how Treatment (providing a more usable system not requiring entering a password continuously) was effective for changing the quality of performance (improved d-prime). Reinforcement withdrawal led to a performance decrement, meaning that the behavior was not immediately generalized. This is only a first attempt to understand whether a token economy can be applied to the cybersecurity setting and should not be considered conclusive. Of course, a research program in this field requires many steps to assess the effect of as many variables as possible.

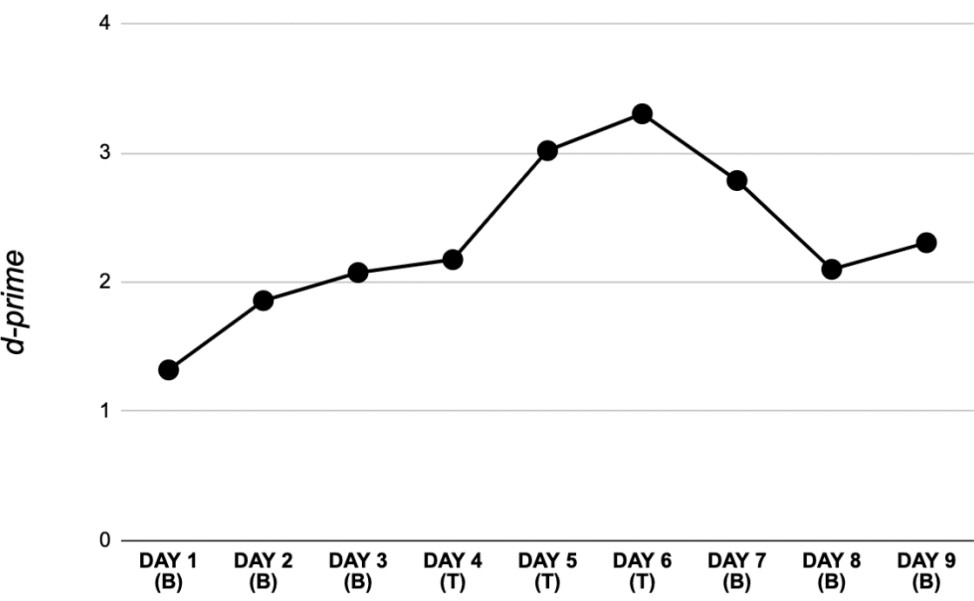

**Figure 1.** Performance of a subject discriminating between suspicious and non-suspicious email with (days 4 to 6) and without (days 1 to 3 and 7 to 9) reinforcement.

## 5. Discussion and Conclusions

The central point of our reflection is that security systems based on the token economy could be implemented by reinforcing secure behaviors, that is, offering usability as a back-up reinforcement for secure behaviors. In practice, this would be a behavior-based security model that exploits the usability/security trade-off instead of reducing the gap between these two needs, an attempt that thus far has not been satisfactorily achieved.

The users would not be merely encouraged to adopt secure behaviors but reinforced for having put them in place. In this case, it is important to remember that a system needs to verify the user's credentials too often during an interaction cannot be defined as very usable. On the other hand, the redundancy of security measures is considered necessary to prevent unaware users from making mistakes. However, a user who provides continued evidence of awareness may "score points" and achieve a "pro" user status, and it may be possible to eliminate some restrictions to ensure greater usability. Would having access

to greater usability improve safety behaviors? No activity can be defined as reinforcing a priori; only the observation of its effects on behaviors can tell us if it is a reinforcer. Therefore, it is essential to test this hypothesis by imagining a research agenda that includes specific experimental activities to answer the following questions:

- Will a reduction in the complexity of the interaction represent a reinforcer for the emission of secure behaviors? The answer to this question is not obvious; the reinforcing stimulus is not based on its intrinsic properties but on the modification of the future probability of emitting the behavior. Therefore, it can be defined only post hoc, based on the effect that the consequence has on the behavior.
- Will the implementation of a token economy system be effective in achieving an increase in secure behavior, in the context of cybersecurity, where the individual's task is to detect suspicious activity during the normal use of technology? The token economy has been used successfully in several fields. There is no reason to rule out that it could show its beneficial effects in the cybersecurity context as well. Of course, this remains to be proven.
- Will the possible beneficial effects of such a program be limited to obtaining tokens, or will they persist after a reinforcement program is completed? In educational contexts in which the token economy has been largely employed, the goal is the generalization of learned behaviors. It is critical to assess whether exposure to a reinforcement program needs follow-up activities to generalize safe behaviors.
- What is the most effective reinforcement schedule to achieve immediate and long-lasting effects? Reinforcement schedules can be based on the amount of behavior produced or the interval required to achieve reinforcement. In addition, they can be fixed or variable. It would be unnecessarily complicated to deepen reinforcement schedules, but it is useful to point out that each type of schedule produces specific effects independently of the organism, the behavior, and the type of reinforcement.
- Will response cost (i.e., punishing insecure behavior) add anything? Reinforcement is a very powerful mechanism, much more than punishment, but the combination of these two strategies is plausible for several practical reasons; encouraging safe driving does not detract from the need to impose fines on those who violate traffic laws.

These questions are just a first set that is essential to answer, but many more could be formulated as knowledge in this area advances. Once the regularities in user behavior have been defined, the next step should be to move away from laboratory research to implement intervention strategies on real platforms. This paper focuses on password management, the authentication method used in almost every web application. Perhaps, systems that do not require passwords may become widespread in the future and allow overcoming the problem of attacks like phishing [57]. However, the research agenda devised here is not aimed at finding technical solutions to be applied to a specific type of system. On the contrary, our aim is to provide a general perspective that could be called "behavior-based cybersecurity" (as in behavior-based safety [58]). In this paper, we wanted to emphasize that it is necessary to start an experimental research program that goes beyond the role to which the human factor is often relegated in the field of cybersecurity: identify differences between types of users almost exclusively using self-reporting questionnaires. Lebek et al. [59], who reviewed psychological approaches to cybersecurity, criticized the use of self-reports and suggested the use of "additional research methodologies such as experiments or case studies" (p. 1063).

The example based on the single case provided above is a starting point. The answer to all our questions is still a long way off.

**Author Contributions:** All authors reviewed the literature, drafted the manuscript, and critically revised and approved the final version before submission. All authors have read and agreed to the published version of the manuscript.

**Funding:** This research received no external funding.

**Institutional Review Board Statement:** Not applicable.

**Informed Consent Statement:** Not applicable.

**Conflicts of Interest:** The authors declare no conflict of interest.

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
