# Peer review of "Getting Rid of the Usability/Security Trade-Off: A Behavioral Approach"

_jcp, doi:10.3390/jcp2020013_

Round 1
Reviewer 1 Report
The manuscript titled “Getting rid of the usability/security trade-off: a behavioral approach”, proposes an approach that incentivizes individuals showing high adherence to security policies with increased usability. It has five sections, among them the first three sections explain concepts of cybercrimes, usability, cybersecurity, and usable security. Section four intends to analyze human behaviors or motivations utilizing the concepts of reinforcement contingency and operant conditioning. Finally, section five presents the authors’ proposition and a list of hypotheses that required further investigation. In total, the manuscript uses 56 references of which 18 are current (from the last five years). The remaining references are old papers but relevant.
It is interesting to see the authors raising an important issue of the “top-down approach” in the human aspect of cybersecurity, for example, policy design. The main issue with the “top-down” approach is that the user’s needs and expectations often take the back seat leading to their incompliance. As a measure, the authors suggest considering the user’s characteristics and ability, even for the usability. It also means that cybersecurity solutions should be more customized, personalized, and localized to meet the user’s needs and situations. Moreover, the approach proposed by the authors has the potential to be a topic for future investigation.
However, there are several serious weaknesses in the manuscript.
- A large part of the manuscript focuses on explaining and clarifying the existing concepts. I did not find the high relevancy of those detailed theories with the main work/proposition of the study. In actuality, the proposition should have received the main attention and provided with better clarification, justification, and if possible, demonstration (with use-case examples) for it.
- The manuscript has nothing on the research methodology followed by the study.
- The authors have often referred to password-based authentication for their work but have completely disregarded “passwordless authentication”. Many existing and emerging passwordless authentication mechanisms (that are both highly secure and usable) are gaining popularity. These mechanisms have a high potential to become the mainstream authentication mechanism in the near future. Either the authors should have explained how their proposition will work for these passwordless authentication mechanisms or have considered another issue in usable security as an example.
- Many (B. Lebek, J. Uffen, M. Neumann, B. Hohler und M. H. Breitner, “Information security awareness and behavior: a theory-based literature review,” Management Research Review, Bd. 37, Nr. 12, November 2014) theories of human behaviors have been tested and implemented, individually or combined, to motivate people’s cybersecurity behaviors. In general, I see the proposed approach as utilizing, intrinsic incentives, to motivate people to act safely. How do the authors differentiate this work from many studies that have utilized intrinsic incentives to motivate security behavioral change?
- More importantly, there exist various drawbacks of using incentives (token economy) in cybersecurity. The incentive-enforced approach may be good for a short-term behavior change but for a long-term behavior change, it may not produce the desired outcome. For example, people start to expect incentives each time they performed the action. Some may feel that the incentive is not as expected or big enough to get motivated and stop getting motivated. Similarly, many may revert back to their earlier behavior as soon the incentive is stopped. How will the proposed approach handle such cases?
- It is also important to investigate “will increasing usability can really motivate people to act?” Because usability is not as simple, clear, and perceptible as other types of incentive, reward, or punishment. It is possible that the users may not notice usability as an incentive or simply take it as an update in the software system. In that case, how the proposed approach will be useful?
- Do the authors have any basis to believe that the proposed intrinsic incentive will affect all types of users? If yes, then need to justify this and if not, it would be better to specify the type of users their work targets. Because there are many influencers like (work) environment, social, economic, and cultural factors that change the user’s course of actions. I believe it would be worthy to discuss the potential impacts of these influencers on the proposed approach.
- Lastly, the authors do not provide any example of how their proposal could be implemented in practice. For example, if we consider the case of password-based authentication, then what would be an “increased usability” mean in this case.
Author Response
[Point 1] and [Point 2]: We are not sure if we understood the criticism here. The only theoretical account we referred to was that provided by behavior analysis, and we have presented it for explaining the idea underlying the research agenda we intend to propose. We think that there is an underlying misunderstanding related to the fact that the manuscript was submitted to the “Viewpoint” category but was erroneously reported as “Research Article” instead. Therefore, this is not an empirical study. It is a documented research perspective based on a specific work hypothesis (to be tested in the future).
[Point 3]: It is true that we have focused on password-based authentication and we are aware of the fact that it is not the only authentication strategy, although it is the most commonly implemented. Our focus on passwords is mainly related to the type of research activity we are devising here, which will provide us with a way for testing our ideas in the future through empirical research. Nevertheless, we agree that it is important to include a reference to this type of approach, and we have added a reference to it in the manuscript.
[Point 4]: We are very grateful to the reviewer for pointing us to the review by Lebek et al. (2014). We had missed that read it and it is now cited in the manuscript. However, none of the studies reviewed there was actually related to the behavior analysis approach (naming something “behavioral”, is not the same thing as being part of the “behavioral analysis” theoretical approach). As we have stated in the conclusions section, “we wanted to emphasize that it is necessary to start an experimental research program that goes beyond the role to which the human factor is often relegated in the field of cybersecurity: identify differences between types of users almost exclusively using self-report questionnaires”. The approaches reviewed by Lebek and colleagues are, indeed, of that type (and they criticize that approach!).
[Point 5]: This is a major question and applies to any area where behavior change is a goal. Generalization is a crucial issue and several procedures have been devised to facilitate maintenance of performance when reinforcement is withdrawn. As we have reported in the manuscript when describing our research agenda “Will the possible beneficial effects of such a program be limited to obtaining tokens, or will they persist after a reinforcement program is completed? In educational contexts in which the token economy has been largely employed, the goal is the generalization of learned behaviors. Assessing whether exposure to a reinforcement program needs follow-up activities to generalize safe behaviors is critical.”
[Point 6]: Very true: we have no idea whether the usability change provided as reinforcement will be “noticed” by the users. That is the reason for this research program. Again, this manuscript was submitted as “Viewpoint” with the aim of proposing research questions and only a trivial mistake (reporting the label “Research Article” in the beginning) led to evaluating it as a research paper providing answers.
[Point 7]: Actually, we think that all types of users could benefit from this approach (assuming that it works in general!), with the only caution of applying it to users that actually need an improvement in their secure behavior. Users that have already good security management skills may not show an increase in their performance within a token economy setting. Nevertheless, the idea of granting them a more usable (or less awkward) system is still a viable perspective. As for the work environment, social, economic, and cultural factors that may change the user’s course of action, it is something important to investigate but not into a laboratory setting as we are proposing here.
[Point 8]: We have added a specific example of how we are going to approach experimentally our hypotheses. Of course, this is not a case study (remember this is a “Viewpoint” article), but it is the first run of an experiment using a single-case research approach.
Reviewer 2 Report
The paper presents an important merge between the psychological factor in human behavior versus the security measures adopted by a computer system. The main contribution of the paper is the idea of using a token system to reinforce complying with security policies. The background for both fields: reinforcement of user behavior by using a token system and the description of cyber-security threats are adequate for the support of the main idea.
The paper needs a case study to support the idea. In the conclusion, the authors propose their idea to be implemented to assess the accuracy of the idea. The reviewer considers that in order for such an idea ot be publishable, it is necessary to have a case study: a design of an experiment that follows the ideas of the paper and a moderate implementation of the experiment. This is the reason for the reviewer recommending a "major revision".
Please note some minor recommendations/comments:
- The first paragraph is the same as the abstract. This is bad practice. It shows the abstract to be watered down. Change the abstract to be more concise on the definitions and more focused on the results and conclusion. The abstract needs to be rewritten.
- The paper contains very good definitions of terms.
- Page 5, last paragraph: "when security solutions are developed, it is important" -- add the comma.
- Page 6, first paragraph: rephrase the sentence starting: "Among the different perspectives in Psychology ... "
- Page 6, the end of the first paragraph: rephrase the comment: "(i.e. in the form that the responses take) "
- Page 8, line 14 from the bottom: "to get rid of some restrictions" -- add the word "of"
- Page 9, line 4: change to "or they will persist after"
- Page 9, line 6: The following sentence has two verbs: "Assessing whether exposure to a 410 reinforcement program needs follow-up activities to generalize safe behaviors is 411 critical."
Author Response
The reviewer requested a case-study. We think that there is an underlying misunderstanding related to the fact that the manuscript was submitted to the “Viewpoint” category but was erroneously reported as “Research Article” instead. Therefore, this is not an empirical study. It is a documented research perspective based on a specific work hypothesis (to be tested in the future). Anyway, we have added a specific example of how we are going to approach experimentally our hypotheses. Of course, this is not a case study (remember this is a “Viewpoint” article), but it is the first run of an experiment using a single-case research approach.
All recommendations kindly provided by the reviewer have been implemented in the manuscript.
Round 2
Reviewer 1 Report
The authors have made the relevant changes to the article. They have satisfactorily fixed the major issue raised, specifically to demonstrate with an example explaining how their proposition could be implemented in practice. The remaining issues have been convincingly answered. I accept the article for publication in JCP.
Reviewer 2 Report
Accept.